# Paclobutrazol Improves the Quality of Tomato Seedlings to Be Resistant to *Alternaria solani* Blight Disease: Biochemical and Histological Perspectives

**DOI:** 10.3390/plants11030425

**Published:** 2022-02-04

**Authors:** Tarek A. Shalaby, Naglaa A. Taha, Dalia I. Taher, Metwaly M. Metwaly, Hossam S. El-Beltagi, Adel A. Rezk, Sherif M. El-Ganainy, Wael F. Shehata, Hassan R. El-Ramady, Yousry A. Bayoumi

**Affiliations:** 1Department of Arid Land Agriculture, College of Agricultural and Food Science, King Faisal University, P.O. Box 400, Al-Ahsa 31982, Saudi Arabia; salganainy@kfu.edu.sa; 2Horticulture Department, Faculty of Agriculture, Kafrelsheikh University, Kafr El-Sheikh 33516, Egypt; yousry.bayoumi@agr.kfs.edu.eg; 3Agricultural Research Center, Plant Pathology Research Institute, Giza 12619, Egypt; naglaa_abdelbaset@yahoo.com (N.A.T.); arazk@kfu.edu.sa (A.A.R.); 4Agricultural Research Center (ARC), Vegetable Crops Research Department, Horticulture Research Institute, Giza 12619, Egypt; daliataher1981@gmail.com; 5Agricultural Botany Department, Faculty of Agriculture, Kafrelsheikh University, Kafr El-Sheikh 33516, Egypt; metwalysalim@yahoo.com; 6Department of Agricultural Biotechnology, College of Agricultural and Food Science, King Faisal University, P.O. Box 400, Al-Ahsa 31982, Saudi Arabia; wshehata@kfu.edu.sa; 7Biochemistry Department, Faculty of Agriculture, Cairo University, Giza 12613, Egypt; 8Plant Production Department, College of Environmental Agricultural Science, El–Arish University, North Sinai 45511, Egypt; 9Soil and Water Department, Faculty of Agriculture, Kafrelsheikh University, Kafr El-Sheikh 33516, Egypt; hassan.elramady@agr.kfs.edu.eg; 10Physiology & Breeding of Horticultural Crops Laboratory, Horticulture Department, Faculty of Agriculture, Kafrelsheikh University, Kafr El-Sheikh 33516, Egypt

**Keywords:** anatomy, biotic stress, catalase, early blight, paclobutrazol, peroxidase, phytopathogens, disease index, antifungal, chlorophyll content

## Abstract

The production and quality of tomato seedlings needs many growth factors and production requirements besides controlling the phytopathogens. Paclobutrazol (PBZ) has benefit applications in improving crop productivity under biotic stress (*Alternaria solani*, the causal agent of early blight disease in tomatoes). In the current study, the foliar application of PBZ, at rates of 25, 50, and 100 mg L^−1^, was evaluated against early blight disease in tomatoes under greenhouse conditions. The roles of PBZ to extend tomato seedling lives and handling in nurseries were also investigated by measuring different the biochemical (leaf enzymes, including catalase and peroxidase) and histological attributes of tomato seedlings. Disease assessment confirmed that PBZ enhanced the quality of tomato seedlings and induced resistance to early blight disease post inoculation, at 7, 14, and 21 days. Higher values in chlorophyll content, enzyme activities, and anatomical features of stem (cuticle thickness) and stomata (numbers and thickness) were recorded, due to applied PBZ. This may support the delay of the transplanting of tomato seedlings without damage. The reason for this extending tomato seedling life may be due to the role of PBZ treatment in producing seedlings to be greener, more compact, and have a better root system. The most obvious finding to emerge from this study is that PBZ has a distinguished impact in ameliorating biotic stress, especially of the early blight disease under greenhouse conditions. Further studies, which consider molecular variables, will be conducted to explore the role of PBZ in more detail.

## 1. Introduction

Tomato (*Solanum lycopersicum* L.) is one of the most valuable vegetable crops worldwide. The Egyptian cultivated area of tomato was 375,276 ha, with a productivity of 38.96 Mg ha^−1^ [1]. The highest Egyptian production of tomatoes during the last decade was 8.6 million metric tons in 2012 [2]. Globally, in 2019, the main producers of tomato included China, which produces alone about 63 million tons, ≈35%, of the total (181 million ton), followed by India, Turkey, the USA, and Egypt producing 19, 12.8, 10.9, and 6.9 million tons, respectively [1]. High seedling quality and their transplantation are mutual practices in the fruitful production of the tomato for fast, sustainable establishing, together with enhancement of earliness, uniform maturity and total yield, as well as quality [3]. The lack of a pre-contracting system for tomato seedlings between the nursery and farmers led the nursery to produce seedlings in a large quantity, sometimes causing a wait for sales [4]. These seedlings may be exposed to damage if they are not transplanted into the field at the appropriate time. Therefore, it is helpful to extend the seedlings life in the nursery, while maintaining high quality without losses. Several compounds can inhibit height growth, hence, extending the life of vegetable seedlings, such as chlormequat chloride (CCC) and daminozide [5], as well as paclobutrazol [6,7,8].

Paclobutrazol (PBZ), a triazole-type plant growth regulator or retardant, is well-known as anti-gibberellins. PBZ can block the conversion of *ent*-kaurene to *ent*-kaurenoic acid during biosynthesis pathway of gibberellin by inhibition of kaurene oxidase [9,10]. Foliar application of Paclobutrazol usually reduces shoot and root length by increasing the stiffness of the cell wall and decreasing cell wall expansion [11]. Several studies confirmed that applied PBZ improved various kinds of compatible solutes and osmo-protectants, such as proline, which increases the plant’s tolerance to water deficits [9,12]. Many benefits of PBZ application have been intensively reported, including improving crop productivity, plant stress tolerance, fruit/grain quality, plant water relation, and membrane stability index [13,14]. In addition, PBZ prevents sucker re-growth in bananas [15], promoting fruit sets in many crops (such as olives) [16], as well as inhibiting the biosynthesis of gibberellin, early fruit set, and reduced stem growth [17]. Concerning the toxicity of PBZ, for living organisms, showed low toxicity via the dermal route in animals, whereas it caused moderate toxicity via human oral and inhalation routes. Based on the available researches, PBZ is considered unlikely to be genotoxic or carcinogenic to humans [18,19].

Fungal diseases are considered one of the core problems facing and affecting tomato seedlings in the nurseries. Among the pathogens that affect tomato seedlings are soil-borne (causing root decay or damping-off) and foliar diseases, including *Alternaria solani* and *Phytophthora infestans*, which reduce yield quality [20]. Among foliar pathogens, *A. solani*, which caused an early blight disease in tomatoes, is a highly destructive pathogen on both open field and greenhouse tomatoes [21,22]. *A. solani* causes infections on foliage, basal stems of transplants, stems of mature plants (stem lesions), and fruits (fruit rot) of tomato [23]. Early blight disease may cause crop losses of up to 78% to solanaceous crops [24].

Regardless of the promising results of chemical treatments in controlling fungal pathogens, phytotoxicity, and chemical residues are major problems that lead to environmental pollution and human health hazards.

Keeping the roles of PBZ under stress in view, the main aim of the present study is (1) to find out whether the foliar application of paclobutrazol has any growth regulatory outcomes on tomato seedlings under both normal and early blight stress disease, (2) to document the extending impact of the applied paclobutrazol handling of tomato seedlings in nurseries, and (3) to observe the biochemical and histological responses of tomato seedlings to paclobutrazol foliar application, under both control and biotic stress conditions.

## 2. Results

### 2.1. Pathogenic Ability of the Four A. solani Isolates on Tomato Plants

An experiment was conducted to assess virulence of four *A. solani* isolates by using a susceptible tomato hybrid (Alissa F_1_) under greenhouse conditions. While all of the four obtained isolates were pathogenic to tomato seedlings, causing identical early blight disease symptoms, isolate number 1 (I_1_) was the greatest virulence isolate in the experiment (Table 1), compared with the other isolates. Isolate I_1_ had the highest disease index percent on tomato plants (23.07, 45.92, and 80.5% after 7, 14, and 21 dpi, respectively). However, the other isolates were varied in their degrees of pathogenicity. Therefore, isolate I_1_ of the pathogen was chosen for the following studies. Differences in the pathogenicity of the tested pathogenic isolates may be due to their physiological and biochemical components. It may also relate to the genetic makeup of host variety and pathogen, as far as their interactions are concerned [25].

### 2.2. Impact of Paclobutrazol on the Linear Growth of A. solani

The in vitro antifungal activity of paclobutrazol showed that all evaluated concentrations of PBZ indicated antifungal activity and significantly inhibited mycelial growth percentage of *A. solani* (Figure 1 and Table 2). All concentrations (25, 50, and 100 mg L^−1^) showed the highest reduction of mycelial radial growth (2, 2.7, and 3 cm), without significant differences in between, suggesting similar potency, compared to control treatment (without PBZ), which resulted no inhibition of mycelia growth (9 cm). The results showed that PBZ had the highest antagonistic effect against of *A. solani* (Figure 1 and Table 2). The reduction percentage of *A. solani*, due to PBZ applications, were the highest values, especially when using the rate of 100 mg L^−1^ (77.8%), without significant differences at 25 and 50 mg L^−1^ rates, 66.7 and 70%, respectively.

### 2.3. Development of Early Blight Disease Due to PBZ Applications

In general, in vitro experiment exogenous application with PBZ obviously reduced the of early blight intensity on tomato leaves, in comparison to the control seedlings, after 7, 14, and 21 days post-inoculation (dpi) (Figure 2 and Table 3). Although, an advanced rise was observed in both the disease incidence and disease severity on control tomato seedlings during the experiment. All PBZ applications significantly decreased the disease incidence (DI) and its severity (DS) percent, at 7, 14, and 21 dpt, until finishing the experiment. Applied PBZ at 25 mg L^−1^ was the most efficient treatment and had the lowermost DI and DS (%) after the previously mentioned periods. It is worth mentioning that PBZ, at all doses, significantly reduced both the DI and DS percent at the three studied stages, and compared with the control treatment. In the same manner, all PBZ applications (25, 50, and 100 mg L^−1^) showed the highest efficacy (87.6, 87.2, and 84.3%, respectively), compared to the untreated plants (Table 3).

### 2.4. Response of Vegetative Growth and Chlorophyll Content to Applied PBZ

To evaluate the response of tomato seedlings to applied doses of PBZ, different vegetative growth parameters, besides chlorophyll content, were measured after 10, 20, and 30 days from PBZ applications (during 2021) (Table 4). The seedlings treated with PBZ, after 10 days, represent the study of the role of applying different doses of PBZ, without infected seedlings and with *A solani* as a control, whereas after 20 and 30 days, as infected seedlings. The commonly known impact of PBZ as a plant growth retarder is clear, due to its decreased seedling height. From Table 4, it can be seen that the seedling height was deceased by increasing the applied doses of PBZ, whereas, in general, seedling height values was increased from 10 to 30 days after foliar-applied PBZ. Although, the increasing rate of seedling height was the highest in control seedlings, as compared with the PBZ application in all stages (10, 20, and 30 days after applications). The applied dose of 50 mg L^−1^ PBZ recorded, in general, the highest values of chlorophyll content, as well as after 10 or 20 or 30 days after foliar applied PBZ; then, the differences were not significant with the control treatment after only 10 days. Stem diameter values increased by an increasing period after PBZ application from 10 to 30, recording the highest value at dose of 100 mg L^−1^ (0.336 mm), compared to control seedlings, which resulted the lowest diameter in all stages. For seedling fresh weight, it was significantly influenced by treatments at all growth stages. After 10 days, the highest values were shown from control seedlings, compared to all PBZ applications. Nonetheless, all doses from PBZ produced the highest values of seedling fresh weight at 20 and 30 days after application, especially at a dose of 100 mg L^−1^. The most obvious observation to appear from the statistics comparison was the dry biomass per seedling, which was significantly differed after 10, 20, and 30 days and the highest values were recorded from applied PBZ at dose of 100 mg L^−1^ (0.37 and 0.62 g, respectively), after 20 and 30 days from application. Root fresh and dry weights were significantly affected by PBZ applications at the three growth stages. PBZ, at a rate of 100 mg L^−1^, resulted highest value of root fresh weight; however, a PBZ dose of 50 mg L^−1^ resulted in the highest value of root dry weight in most cases, compared to the other doses and control.

### 2.5. Response of Enzyme Activities to Applied PBZ

Two plant enzymes (catalase and peroxidase) were evaluated as bioindicators for survival tomato seedlings under biotic stress (Table 5). There was a highly significant relation between applied doses of PBZ and values of the studied enzymes, where the highest applied dose of PBZ (100 mg L^−1^) recorded the highest value of CAT (102.88) and POD (0.057) as mM H_2_O_2_ g^−1^ FW min^-1^, which did not significantly differ with PBZ at a dose of 50 mg L^−1^ in enzyme activities. These results confirmed that the foliar application of PBZ enhanced the cultivated tomato seedlings quality under biotic stress, through promoting and producing higher plant enzymes, which support cultivated seedlings under biotic and abiotic stresses.

### 2.6. Response of Anatomical Features to Applied PBZ

The internal structure of the tomato seedling stem is similar to the other dicotyledonous plants and built-up, essentially, of parenchyma ground tissue, including cortex tissue and pith tissue, regular vascular bundles, and medullary rays, which connect between the cortex and pith tissues. It is clear that, from the present data in Table 6 and Figure 3, the application of paclobutrazol has a positive impact on stem anatomical features, which led to enhancing most of investigated the anatomical measurements of tomato stem, especially the second dose (50 mg L^−1^), which increased the thickness values of the cuticle layer and tissues of epidermis, cortex, xylem, and phloem, as well as the diameter of stem cross-sections and xylem vessels. These obtained results were compared with the control and other concentrations of PBZ used.

Stomata measurements have included stomata density and stomata dimensions (length and width). Data presented in Table 7 and Figure 4 showed that there was an increase in density of stomata, due to the application of PBZ, with various concentrations. These obtained results were compared to the control treatment. This increase of stomata density is due to the negative effect of PBZ on the leaf area, as well as the inhibition of it [26]. The highest values of stomata density were recorded in the application of the second concentration of PBZ, compared with other concentrations. Besides that, using PBZ with investigated concentrations enhanced the values of the stomata dimension (length and width), compared to the control treatment, due to the PBZ application stacking of the stomata, per area unit of leaves.

## 3. Discussion

High-quality tomato seedling production needs to save the required growth factors, including the environmental and practical issues. These issues may include controlling the pests and diseases, as well as proper agricultural practices (e.g., fertilization, irrigation, lighting, etc.). It is very important to produce vigorous tomato seedlings with a long shelf life, especially under intensive work in greenhouses, which sometimes needs a delay in transferring and transplanting seedlings into farming field [27]. This target will be more complicated under conditions of plant diseases, particularly the early blight. To extend the life and juvenility of tomato seedlings, without losses, many substances have been applied, such as chlormequat chloride (CCC), daminozide, and paclobutrazol. The role of applying different doses of PBZ to control the early blight, resulting from *Alternaria solani*, is investigated in the current study. After selecting the most aggressive, isolated inoculant of *Alternaria solani*, which had the highest disease index after 7, 14, and 21 days from the inoculation, different doses of applied PBZ were investigated on disease incidence and its severity of early blight pathogen (*A. solani*) on tomato seedlings under greenhouse conditions (Figure 1 and Figure 2 and Table 1, Table 2 and Table 3).

Many studies published on the role of paclobutrazol in growing tomato seedlings, including applied different doses of PBZ (i.e., 50 and 100 mg L^−1^) through seed treatment or watering seedlings [28], production of tomato seedlings by applying PBZ (50, 100, and 150 mg L^−1^) using two tomato hybrids [29], and accelerating growth of tomato seedlings by applying 25, 50, and 100 mg L^−1^ PBZ [30]; however, to our knowledge, there are no studies on the crucial role of PBZ against early blight (*Alternaria solani*). On the other hand, many studies reported on the reduction of vegetative growth via applied PBZ on many cultivated crops, such as potato [31], mango [32], *Leonotis leonurus* L. [33], and olive [16], as well as its potential under stress, i.e., drought [13,14,34] and salinity [9,35].

In the current study, in response to the question: what is the impact of paclobutrazol on the vegetative growth of tomato seedlings, a range of responses were elicited. The overall response to this question was very positive. The vegetative growth parameters were tested after 10, 20, and 30 days after the applied PBZ doses, where parameters after 10 days were without infected seedlings, but after 20 and 30 days, as infected seedlings. The direct cause of the increase in leaf darkness and greenness of treated seedlings with PBZ may be due to the increase in chlorophyll content. Surprisingly, PBZ was found to decrease the dry weight and chlorophyll content of seedlings after 10 days, but increased by increasing PBZ levels after 20 and 30 days. The increased chlorophyll content could be due to an increase in the activity of oxidative enzymes, which changed in the levels of carotenoids, ascorbate, and ascorbate peroxidase [31]. Plant enzymes, including CAT and POD, were increased by increasing applied doses of PBZ up to 100 mg L^−1^, recording 0.057 and 102.88, respectively (Table 5). This study produced results that corroborate the findings of a great deal of previous work in promoting growth of plants under stress by applying PBZ. PBZ improves plant tolerance against different stresses by increasing proline content and enzymatic antioxidants [14], increases fruit yield (due to the relatively stouter canopy of PBZ-treated plants), improving rooting system (which may increase the uptake of water and nutrients) [36], regulating photosynthetic capacity and delaying leaf senescence [37], improving the resistance against many plant pathogens [38], and acting as a systemic fungicide against several economically fungal diseases [39]. The mode of action of paclobutrazol may include the inhibition of gibberellic acid synthesis in plants, which reduces gibberellins level, slows cell division and elongation (without causing toxicity to cells), and increases cytokinin content, as well as the root activity and C: N ratio. Therefore, PBZ can delay senescence and extend the juvenility of seedlings, which increased the seedlings life without losses; additionally, it increased the resistance against most of pathogens in the nursery. Interestingly, few studies have reported the potential of paclobutrazol in improving the levels of chlorophyll, antioxidants, and proline contents under various biotic and abiotic stresses, as well as extending the plant growth cycle by delaying physiological maturity [40,41,42,43].

All anatomical features of the seedlings were influenced by the different doses of PBZ applied, particularly the stomata measures, including thickness, width, and numbers, which increased by increasing the applied doses of PBZ, up to 100 mg L^−1^ (Table 7). This increase in tomato stem diameter is achieved via the application of paclobutrazol treatment, due to its role to induce an increase in the vascular bundles’ thickness, thicker cortex tissue, and wider pith tissue diameter, which is associated with larger medullary cells [44]. The highest value in the thickness of the stem cuticle (7.55 µm) was achieved by applying 50 mg L^−1^ PBZ, which may support the resistance of tomato seedlings to early blight (Table 6). The highest numbers or values of stomata of numbers (38.33), thickness (46.21 µm), and width (93.01 µm) may lead to an increase in the efficiency of photosynthesis. Similarly, Tekalign and Hammes [31] stated that applying PBZ on potato leaves increased the anatomical parameters, i.e., thickness cortex, pith diameter, and size of the vascular bundles, and resulted in the thickest stems. This might be due to the radial enlargement of cells because of the decreased endogenous gibberellin activities in response to the treatment. In addition, using PBZ with investigated concentrations led to enhanced values of stomata dimension (length and width), compared to the control treatment, due to the fact that PBZ application stacked the stomata per area unit of leaves. This increase of stomata density is due to the negative effect of PBZ on the leaf area, as well as the inhibition of it [26].

## 4. Materials and Methods

### 4.1. Isolation, Purification, and Identification of Causal Organism

Four pathogenic fungi of *A. solani* were isolated from different tomato fields in Kafr El-Sheikh governorate, Egypt. Briefly, tomato plants, showing typical symptoms of early blight disease, were collected from different locations. Infected leaves and stems were washed using tap water and cut into small parts (5 mm), sterilized using solution of 2% sodium hypo chloride for 2–3 min, then washed three times by sterilized distilled water (SDW). They were then dried between two layers of sterilized filter papers to remove excess water and plated onto petri dishes, 9 cm (in diameter), containing 15 mL potato dextrose agar (PDA) medium, amended with 100 mg L^−1^ streptomycin sulphate at 28 ± 2 °C for 7 days [45]. The hyphal tip technique was used to purify the developed fungal cultures. Four isolates were characterized as *A. solani*, based on the morphological characters counting conidia size, number of longitudinal and transverse septa, and length of a beak [46]. Then, the four isolates were assured as *A. solani*, based on their pathogenicity and typical early blight disease symptoms on tomato plants.

### 4.2. Pathogenicity Test

Pathogenicity test of four *A. solani* isolates were confirmed on a highly susceptible hybrid of tomato (Alisa F_1_ hybrid) in pot trials under greenhouse conditions. Shortly, mycelial mats were harvested from seven-day old cultures of *A. solani*, then milled in 100 mL sterilized distilled water using sterilized mortar; it was filtered and put in a test tube, according to [47]. Thereafter, spore suspensions (10^6^ spores mL^−1^) were prepared for each of the four isolates in sterilized water. At 40 days old, tomato transplants were sprayed with tested inoculum of *A. solani* isolates, as spore suspension (30 mL plant^−1^), while control plants were treated with same amount of distilled water. Inoculated plants were kept under polythene bags for 48 h to raise humidity and then incubated under greenhouse conditions. Disease index (DI%) was assessed and results were recorded three times frequently (7, 14, and 21 days post-inoculation (dpi)) to detect development of early blight disease. In this trial, six replicates were used; each replicate contains five pots (20 cm diameter) with two plants in each pot.

### 4.3. Antifungal Activity

In vitro antifungal activities of PBZ were evaluated by the agar diffusion technique [48]. Briefly, three concentrations of PBZ (25, 50, and 100 mg L^−1^), besides the control, were mixed in proper volumes, individually concentrate with 100 mL of the PDA medium, in sterilized Petri dishes to find required concentration. The negative control was sterilized with PDA medium. Then, the pre-prepared Petri dishes were inoculated with 5 mm diameter mycelial mass of freshly prepared culture of the pathogen (I_1_ isolate), incubated at 27 ± 1 °C, and fungal growth was recorded for 7 days post-inoculation (dpi). The whole experiment used six replications for all treatments. The inhibition percentage of the mycelial growth has been determined using this equation:$$Inhibition\;\left(\%\right)=\frac{C-T}{C}\times 100$$
where “*C*” shows the mycelial growth in negative control dish, and “*T*” show the mycelial growth in different treatments.

### 4.4. Plant Materials and Growth Conditions

This study was carried out in a nursery of the Faculty of Agriculture, April 2021, Kafrelsheikh University in Kafr El-Sheikh governorate, to examine the extending of tomato seedlings life using different doses of PBZ. Paclobutrazol was obtained from Shoura Company for chemicals, Cairo, Egypt, as super coltar 25% PBZ. Paclobutrazol was dissolved in water to make solutions of four concentrations that have been used in the present study (i.e., 0, 25, 50, and 100 mg L^−1^). Throughout the current study, tomato genotype (*Solanum lycopericum* L.—Alissa F_1_ hybrid), which was more susceptible to early blight disease, was used as an experiential plant. Tomato seeds were obtained from the Nunhems Netherlands BV Company, Nunhem, Netherlands and sown in seedling trays in the nursery of a protected cultivation center, Faculty of Agriculture, Kafrelsheikh University, Egypt. Styrofoam trays, with 209 compartments, were filled with a mixture of coco peat: vermiculite (1:1 as *v*/*v*). Treatments were arranged in six replicates; each replication was one tray per treatment (209 cells). All trays were planted manually, with 209 seeds per tray, and covered with the above-mentioned media. After sowing, the trays were put in a plastic house with temperatures ranging from 20 to 30 °C. Trays were watered every 2–3 days using a sprinkler system to maintain substrate at field capacity. During the growth of the seedlings, they were fertilized one time in each trial, after over emergence by a soluble compound fertilizer.

At the second true leaf growth stage, seedlings were sprayed with four treatments of paclobutrazol (0, 25, 50, and 100 mg L^−1^ PBZ. After 10 days from PBZ applications, tomato seedlings were sprayed with a spore suspension of *A. solani*. inoculum (10^6^ spores mL^−1^), as 250 mL seedling tray^−1^, whereas the similar amount of distilled water was sprayed on the control seedlings. The most aggressive isolate of *A. solani* (isolate number 1, I_1_) was used in our study. Tomato seedlings were sprayed using a manual pump sprayer, with an appropriate flow rate, until runoff.

Disease incidence (DI) of early blight was evaluated three times after inoculation, at 7, 14, and 21 days post inoculation (dpi). At the 7th, 14th, and 21st dpt, disease severity (DS) of the typical symptoms of early blight disease was assessed [47]. For all treatments, six replications were investigated.

### 4.5. Assessment of Vegetative Growth and Chlorophyll Content

Vegetative growth parameters of tomato seedlings were assessed for all treatments after 10, 20, and 30 days from PBZ application. The growth parameters included seedling height (cm), stem diameter (mm), and fresh and dry weights of seedling and roots (g) per one seedling. Dry mass was measured after drying at 65 °C for 48 h. Total chlorophyll content was recorded in the fully expanded seedling leaf, via the SPAD-501 chlorophyll meter (SPAD-501, Konica Minolta, Tokoyo, Japan), according to [49].

### 4.6. Enzyme Activities

For enzyme analysis, after 10 days from foliar application of PBZ, samples from fresh leaves tissues were used to measure the total soluble enzymes activity of Catalase (CAT) activity, according to [50], and peroxidases (POD), according to [51].

### 4.7. Anatomical Measurements

For anatomical investigation, transverse sections were taken from the tomato seedlings ten days after PBZ application. The selected treatments samples used the second internode of the stem from the apex. The chosen samples were killed and fixed for 48 h in (FAA) solution (10 mL formalin, 5 mL glacial acetic acid, and 85 mL ethyl alcohol 70%), then washed in ethyl alcohol 70% twice. The dehydration of the samples was performed by passing it in a series of concentrations of ethyl alcohol, followed by embedding it in paraffin wax of 54 °C melting point. Sectioning, at a thickness of 12 (μm), was done with a rotary microtome (model Leica RM 2125, Leica company, Wetzler, Germany)), followed by staining with safranin and light green. The samples were cleared in xylene and mounted in Canada balsam, prepared for microscopic examination [52]. Five reading from each slide were examined with electric microscope (Leica DM LS, Wetzler, Germany) and digital camera (Leica DC300, Wetzler, Germany), then photographed. The histological manifestations were calculated using Leica IM 1000 image management software. Leica software was calibrated utilizing a 1 cm stage micrometer, scaled at 100 μm increments (604364 Leitz Wetzler, Germany) at 10× magnification. The chosen sections were examined microscopically to detect histological features to follow the changes occurring in the stems of tomato plants, as affected by the application of three different concentrations of PBZ, i.e., 25, 50, and 100 mg L^−1^. The histological features in the stem sections are the vascular bundle dimensions (thickness and width), xylem vessels diameter, and thickness of xylem and phloem tissue. One developed mature leaf was randomly chosen after 10 days from PBZ application. Upper epidermis imprints were formed from the middle of each leaflet blade using Cyanoacrylate adhesive (Amir Alpha, www.amazon.eg, accessed on 21 January 2022). A drop of the adhesive was placed on a microscopic slide and quickly pressed on the desired spot of a leaflet, baked by hand. After hardening, the adhesive forms replica of the leaf surface; it was gently peeled off, and the slides were kept for microscopic measurements [53]. Each imprint was examined and photographed with an electric microscope with a digital camera; from each photograph, the number of contained stomata were counted in square microns (μm^2^) using the Leica IM 1000 image management software, Wetzler, Germany).

### 4.8. Statistical Analyses

All the obtained results of the experiments were tabularized and statistically analyzed using analysis of variance method, by means of Co-STAT computer software package, IBM, Armonk, NY, USA, and Duncan’s multiple range test was used to compare between means of treatments [54].

## 5. Conclusions

Healthy and vigorous tomato seedlings are necessary for tomato production, where strong seedlings can support plant productivity in the farming field. The extending of seedling life is also considered an important agro-practice in several nurseries, especially under intensive work to avoiding seedlings losses or damages. The current study was carried out to evaluate whether applying different doses of PBZ can enhance the growth and quality of tomato seedlings, as well as suppress the early blight under greenhouse conditions. The results of this research support the idea that PBZ is not only a plant retardant or plant growth regulator but also a stress ameliorant. Applying PBZ enhanced tomato seedlings quality, as the vegetative growth, through the inhibition of stem cell elongation, reduced the length of internodes of the stem, as well as the size and volume of leaves, and increased chlorophyll production. This is the first study on PBZ that examines the associations between applied doses of PBZ on tomato seedling resistance to the early blight pathogen (*A. solani*). Taken together, these findings suggest the role of PBZ in promoting the life of tomato seedlings, with high quality and without losses. The findings of this investigation complement those of earlier studies in the field of tomato seedling production, particularly under different stresses in particular biotic ones. These findings raised important theoretical issues that have a bearing on the environmental dimension of PBZ: are there any ecotoxicological impacts of applied PBZ on the agroecosystem?

## Figures and Tables

**Figure 1 plants-11-00425-f001:**
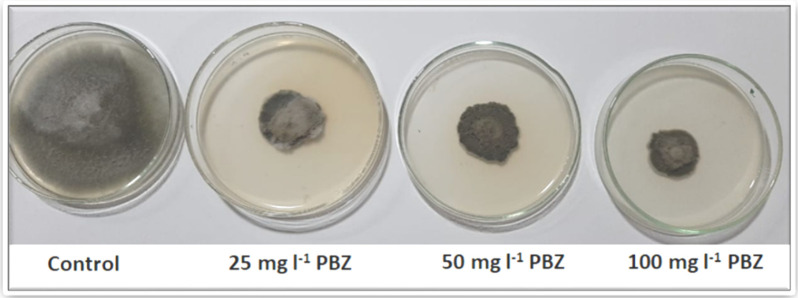
In vitro antifungal activity of paclobutrazol (PBZ) against *Alternaria solani.* The highest applied dose of PBZ (100 mg L^−1^) recorded the highest control of this disease, compared to the control.

**Figure 2 plants-11-00425-f002:**
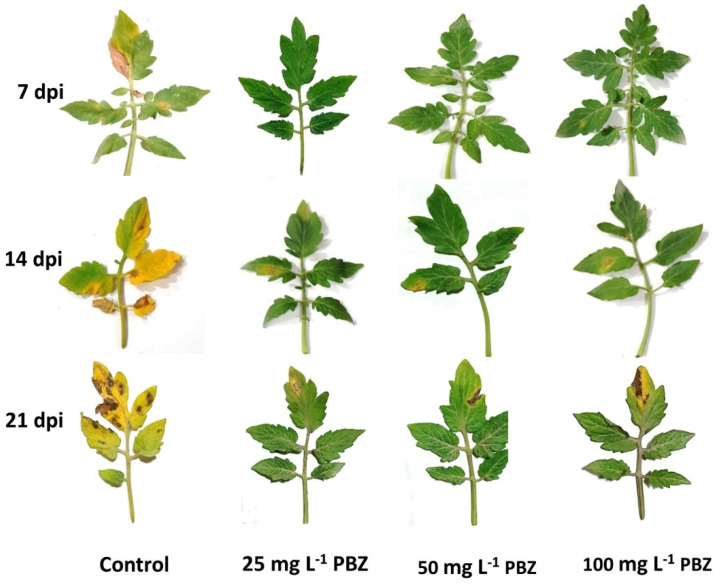
Effect of Paclobutrazol (PBZ) on progress symptoms of early blight disease on tomato seedlings under greenhouse conditions at 7, 14, and 21 dpi (day post inoculation).

**Figure 3 plants-11-00425-f003:**
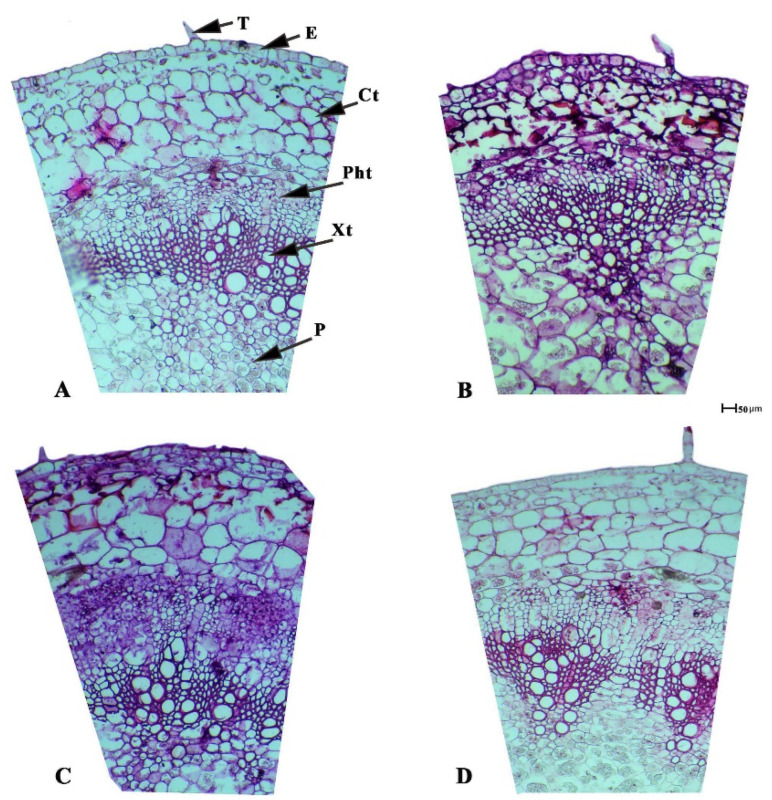
Transverse sections through the tomato stems, as affected by the application of various concentrations of PBZ substance, 10 days after applications, where (**A**–**D**) represent the treatments of the control, applied PBZ at 25, 50, and 100 mg L^−1^. The abbreviations: T (cuticle), E (epidermis), Ct (cortex tissue), Pht (phloem tissue), Xt (xylem tissue), P (pith), and MR (medullary rays).

**Figure 4 plants-11-00425-f004:**
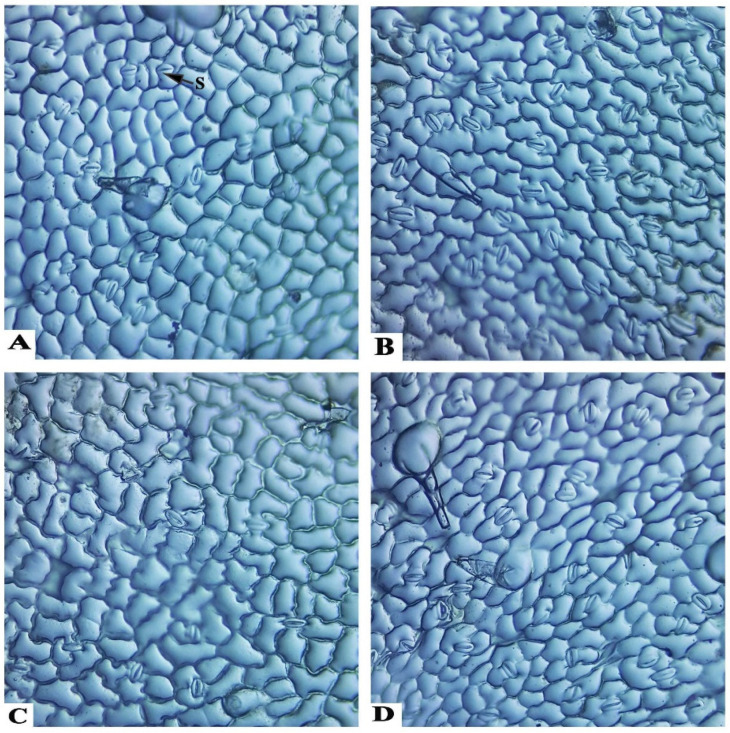
Photographs of stomata on adaxial (upper) surfaces of the tomato leaves, as affected by the application of various concentrations of PBZ substance, 10 days after applications, where (**A**–**D**) represent the treatments of the control, applied PBZ at 25, 50, and 100 mg L^−1^. The abbreviation: S, stomata.

**Table 1 plants-11-00425-t001:** Pathogenic ability of the isolates of *Alternaria solani* on tomato seedlings in pots under greenhouse conditions.

Isolate No.	Disease Index (%)		
	7 Days	14 Days	21 Days
Isolate no. I1	23.05 ± 0.89 a	45.92 ± 2.36 a	80.50 ± 2.39 a
Isolate no. I2	17.07 ± 1.63 b	38.07 ± 1.62 b	54.00 ± 3.26 b
Isolate no. I3	9.80 ± 0.73 d	29.44 ± 1.62 d	40.50 ± 1.24 d
Isolate no. I4	15.60 ± 0.77 c	32.50 ± 1.63 c	45.80 ± 1.67 c
F. test	**	*	**

Where: I1, I2, I3m and I4 are the four isolates of *Alternaria solani*, which were used in the study, Mean values in each column, followed by the same letter, are not significant (*p* < 0.05), * and ** indicating significant and highly significant, respectively.

**Table 2 plants-11-00425-t002:** The inhibition of *Alternaria solani* mycelial growth by PBZ application in different concentrations on PDA medium.

Treatments	Mycelial Growth (cm)	Reduction Rate (%)
Control	9.0 ± 0.0 a	0.01 ± 0.004 b
PBZ 25 mg L^−1^	3.0 ± 0.082 b	66.7 ± 0.735 a
PBZ 50 mg L^−1^	2.7 ± 0.082 b	70.0 ± 2.450 a
PBZ 100 mg L^−1^	2.0 ± 0.082 b	77.8 ± 1.563 a
F. test	**	**

Mean values in each column, followed by the same letter, are not significant (*p* < 0.05). PDA: potato dextrose agar medium. ** indicating highly significant.

**Table 3 plants-11-00425-t003:** Effect of PBZ applications on both disease incidence (DI) and disease severity (DS); % of tomato early blight pathogen (*A. solani*) under greenhouse conditions at 7, 14, and 21 dpi and an efficacy % at 21 dpi.

Treatments	After 7 Days		After 14 Days		After 21 Days		
	DI (%)	DS (%)	DI (%)	DS (%)	DI (%)	DS (%)	Efficacy (%)
Control	50.2 ± 1.95 a	35.4 ± 3.17 a	60.7 ± 5.10 a	50.6 ± 5.47 a	87.9 ± 7.05 a	79.2 ± 4.59 a	00.0 ± 0.0
PBZ 25 mg L^−1^	9.4 ± 3.57 c	3.6 ± 0.95 b	10.2 ± 1.65 c	5.5 ± 1.25 b	18.4 ± 3.18 b	9.8 ± 1.58 b	87.6 ± 5.19
PBZ 50 mg L^−1^	10.3 ± 2.07 c	3.6 ± 0.89 b	18.5 ± 2.57 b	8.1 ± 1.98 b	25.2 ± 1.95 b	10.1 ± 1.99 b	87.2 ± 4.66
PBZ 100 mg L^−1^	15.8 ± 1.85 b	3.6 ± 1.05 b	25.1 ± 3.07 b	10.3 ± 2.05 b	30.1 ± 3.07 b	12.4 ± 2.17 b	84.3 ± 5.05
F. test	**	**	**	**	**	**	-

Mean values in each column, followed by the same letter, are not significant (*p* < 0.05). ** indicating highly significant.

**Table 4 plants-11-00425-t004:** Response of some vegetative growth parameters and chlorophyll content to PBZ doses after 10, 20, and 30 days from PBZ foliar application (during April 2021) (with/without infection by *Alternaria solani*).

Treatments	Seedling Height (cm)	Stem Diameter (mm)	Seedling FW (g)	Seedling DW (g)	Root FW (g)	Root DW (g)	Chlorophyll Content (SPAD)
After 10 days (Not infected seedlings by *Alternaria solani*)							
Control	13.24 ± 1.50 a	0.207 ± 0.021 a	2.65 ± 0.501 a	0.30 ± 0.006 a	0.86 ± 0.14 a	0.085 ± 0.005 a	28.4 ± 2.35 a
PBZ 25 mg L^−1^	8.83 ± 1.68 ab	0.215 ± 0.015 a	1.62 ± 0.452 b	0.25 ± 0.005 ab	0.47 ± 0.05 b	0.071 ± 0.005 a	26.2 ± 2.17 b
PBZ 50 mg L^−1^	6.83 ± 1.29 b	0.238 ± 0.011 a	1.48 ± 0.363 b	0.18 ± 0.003 c	0.44 ± 0.07 b	0.054 ± 0.003 a	27.6 ± 3.02 ab
PBZ 100 mg L^−1^	9.56 ± 1.55 ab	0.266 ± 0.03 a	1.54 ± 0.295 b	0.22 ± 0.005 bc	0.36 ± 0.07 b	0.063 ± 0.003 a	24.3 ± 1.95 c
F. test	**	NS	*	**	*	NS	**
After 20 days (Infected seedlings by *Alternaria solani*)							
Control	14.75 ± 2.17 a	0.275 ± 0.007 a	3.05 ± 0.524 ab	0.33 ± 0.005 b	0.94 ± 0.18 c	0.082 ± 0.005 b	28.8 ± 3.05 c
PBZ 25 mg L^−1^	11.94 ± 1.66 b	0.296 ± 0.025 a	3.12 ± 0.354 ab	0.34 ± 0.007 b	1.02 ± 0.25 b	0.088 ± 0.007 b	31.3 ± 2.84 b
PBZ 50 mg L^−1^	9.94 ± 1.59 b	0.299 ± 0.034 a	2.88 ± 0.441 b	0.29 ± 0.006 c	0.89 ± 0.18 c	0.120 ± 0.008 a	38.6 ± 3.10 a
PBZ 100 mg L^−1^	11.22 ± 2.09 b	0.310 ± 0.033 a	3.23 ± 0.455 a	0.37 ± 0.007 a	1.05 ± 0.20 a	0.089 ± 0.007 b	33.9 ± 1.99 b
F. test	**	NS	*	**	**	**	**
After 30 days (Infected seedlings by *Alternaria solani*)							
Control	27.33 ± 3.17 a	0.254 ± 0.028 b	3.15 ± 0.625 ab	0.42 ± 0.008 b	0.93 ± 0.19 b	0.091 ± 0.009 c	24.9 ± 2.25 c
PBZ 25 mg L^−1^	11.79 ± 1.29 b	0.325 ± 0.033 a	3.42 ± 0.605 ab	0.57 ± 0.009 ab	1.14 ± 0.24 a	0.098 ± 0.009 b	32.1 ± 2.19 b
PBZ 50 mg L^−1^	9.86 ± 1.45 b	0.318 ± 0.029 a	2.96 ± 0.385 b	0.44 ± 0.008 b	0.95 ± 0.23 b	0.137 ± 0.012 a	38.9 ± 3.55 a
PBZ 100 mg L^−1^	11.48 ± 1.88 b	0.336 ± 0.017 a	3.55 ± 0.550 a	0.62 ± 0.009 a	1.22 ± 0.026a	0.132 ± 0.015 a	32.7 ± 3.06 b
F. test	**	**	**	*	**	**	**

Mean values in each column, followed by the same letter, are not significant (*p* < 0.05). Root fresh or dry weights were per one seedling, FW (fresh weight) and DW (dry weight), * and ** indicating significant and highly significant, respectively.

**Table 5 plants-11-00425-t005:** Impact of applied treatments on enzyme activities (catalase, CAT, peroxidase, and POD) after 10 days form PBZ foliar application (after 10 days from PBZ application and without infection by *Alternaria solani*).

Treatments	POD (mM H_2_O_2_ g^−1^ FW min^−1^)	CAT Activity (mM H_2_O_2_ g^−1^ FW min^−1^)
Control	0.012 ± 0.011 ^c^	2.82 ± 0.429 ^c^
PBZ 25 mg L^−1^	0.032 ± 0.020 ^b^	64.87 ± 4. 298 ^b^
PBZ 50 mg L^−1^	0.049 ± 0.029 ^ab^	90.30 ± 5.556 ^a^
PBZ 100 mg L^−1^	0.057 ± 0.025 ^a^	102.88 ± 5.939 ^a^
F. test	**	**

Mean values in each column, followed by the same letter, are not significant (*p* < 0.05). ** indicating highly significant.

**Table 6 plants-11-00425-t006:** Anatomical measurements of tomato stems, as affected by the application of various concentrations of PBZ substance (after 10 days from PBZ application and without infection by *Alternaria solani*).

Anatomical Measurements		Applied PBZ (mg L^−1^) Doses at the Second True Leaf Stage				
		Control	25	50	100	LSD 0.05
Thickness (µm)	Cuticle	4.37 ± 0.69 b	7.24 ± 0.72 a	7.55 ± 0.81 a	5.73 ± 0.78 b	1.42
	Epidermis	33.70 ± 2.09 b	37.84 ± 1.19 a	26.93 ± 0.82 c	27.22 ± 0.48 c	3.46
	Cortex	444.65 ± 14.36 a	304.87 ± 11.20 c	435.64 ± 8.66 a	396.43 ± 34.04 b	37.25
	Xylem	387.53 ± 8.58 c	476.97 ± 30.32 a	444.30 ± 15.09 ab	415.46 ± 10.36 bc	34.31
	Phloem	95.00 ± 4.80 c	88.76 ± 2.50 c	155.48 ± 6.80 a	109.03 ± 6.11 b	10.07
Diameter (µm)	Xylem vessels	60.87 ± 2.00 a	46.47 ± 1.46 a	58.42 ± 2.01 a	53.58 ± 2.25 a	11.70
	Stem	2194.72 ± 162.23 b	2291.66 ± 167.19 b	2477.26 ± 149.38 a	2581.89 ± 138.94 a	70.40

Mean values in each column, followed by the same letter, are not significant (*p* < 0.05).

**Table 7 plants-11-00425-t007:** Stomata measurements of tomato leaves, as affected by PBZ treatments (after 10 days from PBZ application and without infection by *Alternaria solani*).

Treatments	Stomata Measurements		
	Thickness (µm)	Width (µm)	Numbers
Control	42.09 ± 1.83 b	83.66 ± 2.07 c	23.33 ± 1.15 c
PBZ 25 mg L^−1^	38.16 ± 1.60 c	88.89 ± 1.80 b	24.67 ± 0.57 c
PBZ 50 mg L^−1^	46.21 ± 0.65 a	93.01 ± 1.47 a	36.00 ± 1.00 b
PBZ 100 mg L^−1^	41.61 ± 1.68 b	86.14 ± 1.58 bc	38.33 ± 0.57 a
LSD 0.05	2.86	3.29	1.63

Mean values in each column, followed by the same letter, are not significant (*p* < 0.05).

## Data Availability

Not applicable.

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
