# Peer review of "Paclobutrazol Improves the Quality of Tomato Seedlings to Be Resistant to Alternaria solani Blight Disease: Biochemical and Histological Perspectives"

_plants, 2022, doi:10.3390/plants11030425_

Round 1

Reviewer 1 Report

Comments for authors

  1. Line no 85: Please do italics of word Alternaria solani.
  2. Line no 98: Please do italics of word Alternaria solani.
  3. Line no 128: Remove the tab.
  4. Line no 167: Table 5 should be unbold.
  5. How many seeds per replication you have germinate for the taking physiological data?
  6. As authors mentioned that 6 six replication have been taken, as it has been done for all the experiments including physiological, histological and biochemical experiments?
  7. Line no 397: remove the word "according to".
  8. How did you measure the epidermis, cortex etc?
  9. Improve the language.
  10. Otherwise paper is well written and i am pleased to say that it should be accepted with minor revision.

Author Response

Reviewer 1#

Comments and Suggestions for Authors

Line no 85: Please do italics of word Alternaria solani.

Response: done, thanks!

Line no 98: Please do italics of word Alternaria solani.

Response: done, thanks!

Line no 128: Remove the tab.

Response: done, thanks!

Line no 167: Table 5 should be unbold.

Response: done, thanks!

How many seeds per replication you have germinate for the taking physiological data?

Response: done, thanks!

It was mentioned in materials and methods part: 209 seeds per tray line 345

As authors mentioned that 6 six replication have been taken, as it has been done for all the experiments including physiological, histological and biochemical experiments?

Response: done, thanks!

 It was mentioned as 209 seeds per each tray or each replicate

Line no 397: remove the word "according to".

Response: done, thanks!

How did you measure the epidermis, cortex etc?

Response: done, thanks!

 in materials and methods part in starting line 382 as follows:

Five reading from each slide were examined with electric microscope (Leica DM LS, Germany) with digital camera (Leica DC300, Germany), then photographed. The histological manifestations were calculated using Leica IM 1000 image management software. Leica software was calibrated utilizing 1 cm stage micrometer scaled at 100 μm increment (604364 Leitz Wetzler, Germany) at 10 x magnification.

Improve the language.

Response: done, thanks!

Otherwise paper is well written and i am pleased to say that it should be accepted with minor revision. Thanks

Response: thanks!

Reviewer 2 Report

The manuscript entitled “Paclobutrazol Improves the Quality of Tomato Seedlings to be Resistant to Alternaria solani Blight Disease Biochemical and Histological Perspectives”, authored by Shalaby and colleagues, deals with the investigation of the potential effects derived from the foliar application of Paclobutrazol at different rate concentrations (25-100 mg L-1) against early blight disease on tomatoes under greenhouse conditions. The manuscript is very well written, deals with the topics with extreme professionalism, and contains very important information. However, I noticed that different typos are present in the text (for example many units of measurement are incorrectly written). Moreover, I have some small concern regard this manuscript, and I think that before judge it suitable as publication oin Plants, small changes need to be made.

Despite this reviewer is well aware of the difficulty of writing an abstract with a very limited number of words, this section should be a single paragraph, of a maximum of 200 words. The proposed abstract far exceeds the guidelines provided by the journal. Consequently, I would recommend limiting and shortening this section following the Journal's guidelines.

The number and type of keywords used in the manuscript should be implemented. These words are important to facilitate the research of the manuscript after publication using the worldwide search engines (for example Pubmed, Google Scholar, etc ...). I would suggest authors to add a few more keywords, considering that the maximum number is 10.

In Figure 2, authors should replace the letters A, B and C with the days since infection. I think that way it's much clearer.

In the tables, authors should report not only the medial value, but also the standard deviation. In general, the different letters assigned after statistical analysis should be place as superscript. Furthermore, the caption of each table should clearly report the statistical test through which the assignment of the letters was possible. Finally, the authors should report the hsd post hoc tests as a supplementary file, in order to be able to verify the truthfulness of the statistical analysis.

The materials and methods part should be implemented with useful information to replace the various assays. For example, no information was entered for enzymatic activities.

Regarding of enzymatic activities, why did the authors limit the analysis to the mensuration of POD and CAT when other enzymes are involved in the oxidative response to abiotic and biotic threats?

Line 208 – 209 should be placed as caption of Figure 3 and not as main text.

Author Response

Reviewer 2#

The manuscript entitled “Paclobutrazol Improves the Quality of Tomato Seedlings to be Resistant to Alternaria solani Blight Disease Biochemical and Histological Perspectives”, authored by Shalaby and colleagues, deals with the investigation of the potential effects derived from the foliar application of Paclobutrazol at different rate concentrations (25-100 mg L-1) against early blight disease on tomatoes under greenhouse conditions. The manuscript is very well written, deals with the topics with extreme professionalism, and contains very important information. However, I noticed that different typos are present in the text (for example many units of measurement are incorrectly written). Moreover, I have some small concern regard this manuscript, and I think that before judge it suitable as publication in Plants, small changes need to be made.

Response: done, thanks!

Despite this reviewer is well aware of the difficulty of writing an abstract with a very limited number of words, this section should be a single paragraph, of a maximum of 200 words. The proposed abstract far exceeds the guidelines provided by the journal. Consequently, I would recommend limiting and shortening this section following the Journal's guidelines.

Response: thanks, we followed the instruction of the journal!

The number and type of keywords used in the manuscript should be implemented. These words are important to facilitate the research of the manuscript after publication using the worldwide search engines (for example Pubmed, Google Scholar, etc ...). I would suggest authors to add a few more keywords, considering that the maximum number is 10.

Response: thanks! We added the following words:

Phytopathogens, Disease index, Antifungal, Chlorophyll content

In Figure 2, authors should replace the letters A, B and C with the days since infection. I think that way it's much clearer.

Response: done, thanks!

In the tables, authors should report not only the medial value, but also the standard deviation. In general, the different letters assigned after statistical analysis should be place as superscript. Furthermore, the caption of each table should clearly report the statistical test through which the assignment of the letters was possible. Finally, the authors should report the hsd post hoc tests as a supplementary file, in order to be able to verify the truthfulness of the statistical analysis.

Response: done, thanks!

The materials and methods part should be implemented with useful information to replace the various assays. For example, no information was entered for enzymatic activities.

Response:  thanks!

This because these methods of measuring enzymes already mentioned in details in may publications for us before, but this phytopathogenic need more details!

Regarding of enzymatic activities, why did the authors limit the analysis to the mensuration of POD and CAT when other enzymes are involved in the oxidative response to abiotic and biotic threats?

Response: thanks!

Because these enzymes are the most common measurements to express on biotic or abiotic stress!

Line 208 – 209 should be placed as caption of Figure 3 and not as main text.

Response: done, thanks!

Reviewer 3 Report

I recommend the publication of the article because scientific experimentation concerning the use of new methods for defence against biotic and abiotic diseases is topical and of particular interest.
The aim and objectives of the article have been stated and are very interesting. The use of Paclobutrazol (PBZ) for plant defence is an important topic especially for the development of new methods of plant protection, especially from biotic factors such as fungal diseases. The work carried out is certainly of international interest and the format applied is certainly suitable for a research article. The work is original, of particular interest and can certainly stimulate research on this topic. The length of the article is good for the journal and the graphs and tables are clear and easy to understand. The conclusion summarises the objectives of the work and future prospects.
I suggest publishing the article because the topic of studying new defence methods for protecting plants from fungal diseases is of particular interest, especially with regard to the reduction of synthetic pesticides in agriculture. I find the article clear and easily understandable for the reader.

Author Response

Reviewer 3#

I recommend the publication of the article because scientific experimentation concerning the use of new methods for defence against biotic and abiotic diseases is topical and of particular interest.

The aim and objectives of the article have been stated and are very interesting. The use of Paclobutrazol (PBZ) for plant defence is an important topic especially for the development of new methods of plant protection, especially from biotic factors such as fungal diseases. The work carried out is certainly of international interest and the format applied is certainly suitable for a research article. The work is original, of particular interest and can certainly stimulate research on this topic. The length of the article is good for the journal and the graphs and tables are clear and easy to understand. The conclusion summarises the objectives of the work and future prospects.

I suggest publishing the article because the topic of studying new defence methods for protecting plants from fungal diseases is of particular interest, especially with regard to the reduction of synthetic pesticides in agriculture. I find the article clear and easily understandable for the reader.

Response: done, thanks!
